# Useful Role of a New Generation of Dexamethasone, Vitamin E and Human Serum Albumin Microparticles in the Prevention of Excitotoxicity Injury in Retinal Ocular Diseases

**DOI:** 10.3390/pharmaceutics16030406

**Published:** 2024-03-15

**Authors:** Javier Rodríguez Villanueva, Pedro de la Villa, Rocío Herrero-Vanrell, Irene Bravo-Osuna, Manuel Guzmán-Navarro

**Affiliations:** 1Unidad Docente de Farmacia y Tecnología Farmacéutica, Departamento de Ciencias Biomédicas, Facultad de Farmacia, Universidad de Alcalá, Ctra. de Madrid-Barcelona A-2, Km. 33,600, 28871 Alcalá de Henares, Spain; 2Departamento de Biología de Sistemas, Facultad de Medicina, Universidad de Alcalá, Ctra. de Madrid-Barcelona A-2, Km. 33,600, 28871 Alcalá de Henares, Spain; pedro.villa@uah.es; 3Instituto Ramón y Cajal de Investigación Sanitaria (IRYCIS), Ctra. Fuencarral-El Pardo, Km. 9.100, 28034 Madrid, Spain; 4Departamento de Farmacia Galénica y Tecnología Alimentaria, Universidad Complutense de Madrid, Plaza de Ramón y Cajal, 28040 Madrid, Spain; rociohv@farm.ucm.es (R.H.-V.); ibravo@farm.ucm.es (I.B.-O.)

**Keywords:** microparticles, dexamethasone, excitotoxicity, ocular rat model, electroretinography, immunohistochemistry

## Abstract

Excitotoxicity has been linked to the pathogenesis of several serious degenerative ocular diseases. Long-term overactivation of the NMDA receptor by glutamate in retinal ganglion cells (RGCs) results in degeneration, apoptosis and loss of function leading to blindness. NMDA receptor antagonists have been proposed as a pharmacological blockage of glutamate excitotoxicity. However, an inhibition of the pathway activated by glutamate receptors has intolerable side effects. An interesting pharmacological alternative would be the use of antiapoptotic compounds as RGCs’ neuroprotective active substances. Several mechanisms have been proposed to explain neuroprotection, including anti-inflammatory and scavenging activities. Here, the role of dexamethasone in neuroprotection was studied. For this purpose, original controlled release systems composed of microparticles containing dexamethasone with or without vitamin E and human serum albumin (HSA) were designed. The particles were prepared by the solid-in-oil-in-water (S/O/W) emulsion–evaporation technique. After properly characterization of the particles, they were intravitreally injected into an rat model of acute ocular excitotoxicity injury. The functionality of the retina was determined by electroretinography and RGCs were counted after cell immunohistochemistry. These microparticulate systems showed the ability to maintain normal electroretinal activity and promoted significant protection of RGCs. Through this proof of concept, we demonstrated that dexamethasone could be a useful anti-inflammatory agent to avoid the progression of degenerative ocular diseases. Furthermore, when administered in controlled release systems that provide low concentrations during prolonged periods of time, not only can the patient’s comfort be increased but the cytotoxicity of the drugs can also be avoided.

## 1. Introduction

Excitotoxicity has emerged as a significant factor in the development of various severe degenerative ocular conditions and injuries, including retinal ischemia, traumatic injury, glaucoma and diabetic retinopathy, all of which stand as leading causes of global blindness [1,2]. Addressing the challenge of arresting or mitigating vision loss caused by the progression of these retinal neurodegenerative diseases remains a primary concern in the field [3,4].

Excessive glutamate, known to be neurotoxic, stimulates glutamate receptors, particularly the N-methyl-D-aspartic acid (NMDA) receptor subtype, triggering a substantial influx of Ca^2+^ through NMDA receptor-operated channels [5]. This heightened intracellular Ca^2+^ concentration is predominantly implicated in neuronal excitotoxicity processes and is considered a key mechanism underlying neuronal cell death [6,7].

Immunohistochemical localization of NMDA receptor subunits in the rat retina has identified their presence on retinal ganglion cells (RGCs), as well as displaced amacrine cells in the ganglion cell layer and a subset of amacrine cells in the inner nuclear layer [8]. However, there is no evidence of functional NMDA receptors in photoreceptor cells [9]. Consequently, long-term NMDA overactivation results primarily in the degeneration of amacrine cells and RGCs, but this does not directly affect other cells in the retina such as photoreceptor cells [10]. Thus, NMDA has been commonly used as an experimental tool to induce glutamate-induced excitotoxic RGC death. Upon injecting NMDA (10 mM) into the vitreous chamber of Sprague Dawley rats, a pronounced and acute decline in the number of retinal ganglion cells (RGCs) was observed through immunohistochemistry employing Brn3a and RBPMS antibodies. This decrease reached its maximum, reflecting peak cell death, 48 h after the administration of NMDA [11].

NMDA receptor antagonists have been proposed as a pharmacological approach for the treatment of degenerative diseases where excitotoxicity is involved. However, it is important to note that NMDA receptors play a pivotal role in mediating synaptic transmission and plasticity, with their activity intricately linked to the transcriptional control of glutathione biosynthesis, a key factor in maintaining cellular redox balance [12]. The prospect of completely blocking NMDA receptors raises concerns about potential intolerable side effects. A compelling alternative approach involves exploring the use of antiapoptotic compounds as neuroprotectants for retinal ganglion cells (RGCs).

Several mechanisms have been proposed to explain neuroprotection, including anti- inflammatory and scavenging activities or a lowering of lipid peroxidation [13,14,15]. It has been recently reported that the knockdown of endogenous mitochondrial Ca^2+^ uniporter expression reduces NMDA-induced increases in mitochondrial [Ca^2+^], resulting in resistance to excitotoxicity [16]. A series of non-steroidal anti-inflammatory drugs containing carboxylic groups bound to aromatic rings (salicylate, sulindac, indomethacin, ibuprofen and R-flurbiprofen) have provided evidence that that there is a partial depolarization of mitochondria and inhibition of NMDA-induced mitochondrial Ca^2+^ uptake without affecting [Ca^2+^]_cyt_ increases. Also, these compounds, at very low concentrations, prevented NMDA-induced apoptosis in aged neurons [17].

Dexamethasone (Dxm, 9-fluoro-11,17-dihydroxy-17-(2-hydroxyacetyl)-10,13,16-trimethyl-dodecahydro-cyclopentaphenanthren-3-one), a synthetic fluorinated glucocorticoid, is a homologue of hydrocortisone, with an accessible hydroxyacetyl group and anti-inflammatory action; it is used, among others, as a treatment of choice in diseases that involve ocular inflammation [18,19,20]. Based on the abovementioned information and following previous studies by our research team [21], Dxm’s role in neuroprotection was investigated.

Various nano- and micro-particulate systems and platforms have been suggested to enhance drug bioavailability through intravitreal administration. The use of both technologies allows for the tailoring of personalized therapies by adjusting the formulation amount based on the required drug dose. However, microspheres, with their higher drug-loading capacity compared to nanospheres and their ability to facilitate sustained drug release over an extended period, have emerged as being particularly suitable for addressing chronic ocular conditions [18,21,22].

For this reason, biodegradable microspheres that can be administered intravitreally by conventional needles with high Dxm encapsulation efficiency and good controlled release and cellular behavior are being developed. To enhance the pharmacological, technological and biopharmaceutical properties, vitamin E and human serum albumin (HSA) were included as additives [21]. In this study, to assess the role of microsphere (MS) formulations in ocular excitotoxicity, MSs were administered to a Sprague Dawley rats, which were further treated with intraocular injections of NMDA and kainic acid (KA), as a model of ocular excitotoxicity. KA was used here in order to potentiate the deleterious effect of NMDA. Electroretinograms and immunohistochemical assays were performed to evaluate the functional damage and cell death.

## 2. Materials and Methods

### 2.1. Microspheres

Previously, three biodegradable MS formulations containing dexamethasone (purchased from Sigma-Aldrich, Madrid, Spain) as the active pharmaceutical ingredient were prepared using a solvent-extraction–evaporation method [21]. Poly(D,L-lactide-*co*-glycolide) acid (PLGA, 50:50) Resomer ^®^ RG 503, Mw 35,000 g/mol, inherent viscosity 0.32 dL/g was purchased from Evonik (Essen, Germany) was selected as the MS polymer due to its biocompatibility and biodegradability. Vitamin E (purchased from Sigma-Aldrich, Madrid, Spain) exerts technological and pharmaceutical functions. The inclusion of oily additives, such as vitamin E, in MPs can improve encapsulation efficiency of the active substance (dexamethasone). Vitamin E has also demonstrated the ability to protect against bioactivity loss of microencapsulated proteins during manufacturing [23]. Vitamin E has antioxidant properties and can be useful in neural protection leading to a reduction in the reactive oxygen species produced in excitotoxicity process. Human serum albumin (HSA, purchased from Tebu-bio, Barcelona, Spain) was included in solid state in one of the formulations (F3-MSs) to modulate dexamethasone release (Table 1).

The microspheres were prepared using a solid-in-oil-in-water emulsion extraction–evaporation technique. Briefly, dexamethasone was added to a PLGA solution in dichloromethane to obtain a suspension, which was emulsified with a 2% PVA solution (5000 rpm, 1 min) to form the S/O/W system. In F2, 20 μL of vit. E was added to the dexamethasone/PLGA suspension prior to homogenization while in F3, HSA was first dispersed in the oil by sonication at a low temperature before it was added to the suspension. MS maturation and consolidation was achieved by magnetic stirring for 3 h in a PVA solution (0.1% *w*/*v*). Finally, the particles were filtered through nylon membranes (40, 20 and 2 μm), frozen (−80 °C) and lyophilized. Due to their adequate control release profile, the 20–40 μm MS fraction was chosen for use in the study. Table 1 shows a summary of the main characteristics (composition, drug and human serum albumin loading, burst effect and dexamethasone released after 7 days) for each MS formulation. All MS formulations were easily suspended in hyaluronic acid (0.1% *w*/*v*) and were injected using 32 G needles.

### 2.2. Animals and Treatments

The experimental procedures were carried out in strict accordance with the current regulations for the use of laboratory animals (Association for Research in Vision and Ophthalmology statement for the use of animals in ophthalmic and visual research; Guide for the Care and Use of Laboratory Animals; and European Directive 2010/63/UE), and all efforts were made to minimize animal suffering and number. The University of Alcalá Research Ethics Committee approved the protocol. Sprague Dawley rats, aged 3–4 months, obtained from Harlan Laboratories (Indianapolis, IN, USA) were used in this study. Thirty-six rats were randomly distributed into six groups (Table 2) and housed one to three rats per cage. The animals were bred at the University of Alcalá animal facilities and reared in an artificial 12 h light/dark temperature-controlled cycle with food and water provided ad libitum.

#### 2.2.1. Microsphere Assessment and Intravitreal Injection

The amount of microspheres administered was calculated based on the minimum concentration required for anti-inflammatory activity in vitreous humor (C_ss_ of 0.1 μg/μL), hypothesizing that this concentration ensures an anti-excitotoxic effect in 0.01 mL of rat vitreous humor with a vitreous Dxm elimination half-life (t_1/2_) of 5.5 h (Chang-Lin JE et al., 2011 [24]). According to the abovementioned information and taking into account previous observations, in vitro release data and preliminary studies (see Table 1), it was estimated that 25 μg of F3 MSs in a single injection could deliver an adequate amount of Dxm in the first seven days by diffusion and some erosion. As 4 μL is the maximum recommended volume that should be administered in the posterior segment of rat eyes, a 6.8 mg MSs/mL hyaluronic acid (0.1%) in phosphate buffer suspension of each formulation was prepared to achieve the proper dosage.

In the case of microspheres, careful consideration must be given to their injectability to achieve an administration system that is as minimally invasive as possible for the eye. For this purpose, particles with sizes between 20 and 40 μm were found to be highly suitable [25]. Additionally, the incorporation of pseudoplastic polymers, such as hyaluronic acid, in the dispersion vehicle for the microspheres can prevent needle blockage and improve injectability. All MS suspensions possessed good injectability properties (≤12N over 10 s) through 30- and 32-gauge needles. As a result, this enabled their use without surgical intervention, causing no or reduced damage. Using a 30-gauge needle, an initial incision was made in the dorso-temporal sclera, situated approximately 1 mm from the sclerocorneal limbus in adult rats aged 12–16 weeks. Subsequent to this, a 32-gauge needle, connected to a Hamilton syringe, was inserted into the vitreous cavity. A precise volume of 4 μL of the pertinent MS dispersion in 0.1% (*w*/*v*) hyaluronic acid was then administered into the right eye of each animal. The cannula was retained in position for one minute before being gradually withdrawn. Post-procedure, the animals were housed individually in cages and allowed to recuperate from anesthesia on a warm water pad, with additional provision of ocular lubrication. In group 1, only 4 μL of 0.1% (*w*/*v*) hyaluronic acid was administered. In group 2, 4 μL of a 10 μg Dxm/μL hyaluronic acid 0.1% (*w*/*v*) solution was injected.

After seven days, 4 μL of a NMDA and KA solution was injected into the same eye of each animal as described above. N-methyl-D-aspartate acid and kalic acid (purchased from Sigma-Aldrich, Madrid, Spain) were dissolved in a phosphate-buffered saline solution (PBS), pH 7.4, to a final concentration of 1 mM and 0.3 mM, respectively, and sterile-filtered prior to administration. After that, 2% Methocel (Ciba Vision AG, 8442 Hetlingen, Switzerland) was applied topically to the cornea to prevent corneal desiccation. Animals with lens damage or vitreal hemorrhage were excluded from the study.

#### 2.2.2. Electroretinogram (ERG) Recording

The protocol for recording ERGs was carried out following the previously published protocol by our research group [10,11]. All manipulation procedures were performed under a dim red light (λ > 600 nm) in absolute darkness. On the seventh day of treatment with the MS formulations, ERG recordings were initiated, as depicted in Figure 1. A follow-up assessment of ERG responses was conducted on the fourth day after inducing retinal damage. In each instance, recordings were performed with a minimum of a 24 h gap post-administration to mitigate any potential influence from stress or acute effects stemming from the treatment on the ERG responses.

A brief description of the complete procedure [10] is as follows. Rats kept in darkness for 12 h overnight were subjected to anesthesia through an intraperitoneal (i.p.) injection of ketamine (70 mg/Kg, Ketolar ^®^ 5% Pfizer, Alcobendas, Madrid, Spain) and xylazine (7 mg/Kg, Rompun ^®^ 2% Bayer, Kiel, Germany) in 0.1 mL sterile saline (NaCl, 0.9%). The animals were maintained on a thermal blanket set at 38 °C throughout the entire procedure. Pupil dilation was induced using a topical application of 1% tropicamide (Alcon Cusí, Barcelona, Spain). To prevent corneal surface dehydration and enhance electrical contact with the recording electrodes, 0.2% polyacrylic acid carbomer Methocel (Ciba Vision AG, 8442 Hetlingen, Switzerland) was instilled into each eye. The recording electrodes utilized were DTL fiber electrodes equipped with a silver-coated nylon conductive yarn (X-Static; Sauquoit Industries, Scranton, PA, USA). A 25-gauge platinum needle, placed under the scalp between the eyes, served as the reference electrode, while a gold plate in the mouth functioned as the ground electrode. The anesthetized rats were positioned on a warming table inside a Faraday cage for the entire experimental procedure.

Scotopic flash-induced ERG responses were recorded simultaneously from both eyes in response to light stimuli, provided by a Ganzfeld dome light source, at 15 different increasing intensities ranging from 10^−5^ to 10^0^ cd·s·m^−2^, ensuring the uniform illumination of the retina. At least 15 consecutive response recordings were averaged for each light presentation. Scotopic threshold responses (STRs) were obtained for flash intensities ranging from −5.5 to −4.4-log cd·s·m^−2^. Rod and mixed mediated responses were obtained for flash intensities ranging from −4.0 to 1.5 log cd·s·m^−2^. The interval between light flashes was 10 s for dim flashes (−5.5 to −1.5 log cd·s·m^−2^) and up to 20 s for the highest intensity (−1,2 to 1.5 log cd·s·m^−2^), ensuring a certain response recovery. The ERG signals underwent amplification and band-pass filtering (0.1–1000 Hz) through a DAM 50 commercial amplifier (World Precision Instruments, Aston, UK). Digitalization of the signals was carried out at a sampling rate of 4 kHz using a PowerLab acquisition device (AD Instruments; Oxfordshire, UK). For the visualization of oscillatory potentials, a targeted filtering process was implemented, narrowing the signal bandwidth to the range of 100 to 10,000 Hz. This refinement in signal processing facilitated a more precise and detailed examination of the oscillatory potentials within the electroretinogram data.

Light stimuli were calibrated periodically with a photometer (Mavo Monitor USB, Gossen, Nürenberg, Germany). The recordings were analyzed with the normalization criteria established by the International Society for Clinical Electrophysiology of Vision (ISCEV) for the measurement of the amplitude and implicit time of the different waves studied. According to these, the first scotopic ERG response that was recorded was called the STR and consisted of a positive potential followed by a negative component, known as the pSTR and nSTR, respectively. In evaluating the electroretinogram signals, the amplitude of the pSTR was computed by measuring the distance from the baseline to the peak of the pSTR, occurring approximately 115 ms after the stimulus. On the other hand, the amplitude of the nSTR was determined by gauging the distance from the baseline to the trough of the nSTR, typically observed around 220 ms post-stimulus.

Following the recording of the STR, the rod response was recorded. The rod response consisted of a positive deflection in the ERG; the amplitude of the b-wave of the rod response was measured from the baseline to the hill of the waveform response. Mixed responses were also recorded. Mixed responses consisted of a negative deflection (a-wave) followed by a positive deflection (b-wave). In the assessment of electroretinogram signals, the a-wave amplitude was calculated by measuring the distance from the baseline to the trough of the a-wave, typically occurring approximately 15 ms after the stimulus. On the other hand, the b-wave amplitude was determined by measuring the distance from the trough of the a-wave to the peak of the b-wave, usually observed around 50 ms after the stimulus. In the analysis of the oscillatory potentials, particular attention was given to the maximum peak-to-trough amplitude. ERG wave amplitudes and oscillatory potentials were calculated for each animal group, and the percentage difference between the treated eyes and the control eyes was obtained for each stimulus and was further averaged (mean ± standard deviation).

#### 2.2.3. Immunohistochemistry

Based on the methodology set up by the Pedro de la Villa research group [10,11,17], at the end of the treatment, the animals were euthanized by carbon dioxide (CO_2_) and the retinal tissue was harvested and processed for immunohistochemistry. Before eye enucleation, a suture was placed on the superior pole of each eye to maintain retinal orientation. Enucleated eyes were fixed in freshly made 4% (*w*/*v*) paraformaldehyde in 0.1 M-phosphate-buffered solution (PBS), pH 7.4, for 2 h at room temperature, and rinsed several times in 33% sucrose. Then, the cornea and lens and vitreous body were carefully removed and the retina was dissected out, embedded in tissue freezing medium and frozen with liquid N_2_. Vertical sections of the retinas (≈10 μm thick) were collected from the central retinal regions.

The retinas were subjected to a 72 h incubation period at 4 °C, during which they were immersed in a solution of goat polyclonal anti-Brn3a antibody (1:500; #sc-31984L, Santa Cruz Biotechnology Inc., Santa Cruz, CA, USA). This antibody solution was prepared in 0.1 M PBS supplemented with 1% (*v*/*v*) Triton X-100 (Sigma), anti-ZNP-1 antibody (1:50; Abcam, Cambridge, UK), anti-Car, bOps and Chat antibodies (1:500, 1:10,000 and 1:1000, respectively; MyBioSource, San Diego, CA, USA) and anti-DAPI antibody to stain the cell nuclei (1:1000; Merck KGaA, Darmstadt, Germany). Then, the retinas were washed, flat-mounted on glass slides with the vitreous side up, coverslipped with an antifading mounting medium (Citifluor Ltd., London, UK) and sealed with nail polish.

#### 2.2.4. Confocal Microscopy and Quantification of Surviving RGCs

Using a laser-scanning confocal microscope (TCS SP2, Leica Microsystems, Wetzlar, Germany), serial horizontal xy-sections, 4 μm in depth, were acquired in the *z*-axis with a 20X objective along the dorsal–ventral and nasal–temporal axes of the retina. RGCs were labeled with antibodies against a specific RGC marker, the transcription factor Brn3a, and antibodies against ZNP-1, Car, bOps and Chat. RGCs in the ganglion cell layer were scored in maximal confocal projections in 8 regions of interest (Figure 2 shows the areas in each retinal quadrant at different eccentricities, 2 and 4 mm from the optic disc, measuring 400 × 400 μm^2^ each). Mean density (number of cells per mm^2^) values were calculated for all eccentricities as well as over the whole retina. A total of 4 retinas per experimental group (NMDA/KA, vehicle, Dxm-vehicle, F1-MSs, F2-MSs and F3-MSs in vehicle) were analyzed.

### 2.3. Statistical Analysis

Previously [21], the statistical significance of the different parameters (yield of product, loading efficiency and particle size) among different batches of microspheres was tested by a one-way analysis of variance (ANOVA) with the Pairwise Multiple Comparison Procedures. Dxm-loaded microspheres’ release profiles were compared with the similarity factor f_2_. Differences were considered to be significant at a level of *p* < 0.05.

Here, statistical analyses were performed using SPSS 18.0 software (IBM Armonk, NY, USA) and GraphPad Instat ^^®^^ 3 for Windows ^^®^^ (GraphPad Software, San Diego, CA, USA). Descriptive statistics were calculated, and the normality of the distribution of the data was examined. To assess the impact of the treatment on ERG responses, a two-way repeated-measures ANOVA was executed, comparing the effects of F1-F3-MSs + NMDA/KA against Dxm, Blank-MSs, Vehicle + NMDA/KA or NMDA/KA alone. Additionally, a two-way ANOVA was employed to scrutinize differences in the mean density of retinal ganglion cells (RGCs) across various experimental groups (NMDA/KA, Vehicle, Blank-MSs, F1-MSs, F2-MSs, and F3-MSs) at distinct eccentricities. In instances where a significance level of 0.05 was attained, post hoc pairwise comparisons were conducted using Bonferroni’s test. Statistically significant results were established for *p* values below 0.05. The presentation of data in plots followed the format of mean ± standard deviation (mean ± SD). When needed, the coefficient of variation (CV) is presented as a percentage SD/arithmetic mean ratio. Microsoft Excel (2019) were used to create the bar graphs. ANOVA tests were used to assess the significant differences among the various experimental groups.

### 2.4. Limitations

In this study, the ability of PLGA microspheres loaded with Dxm to prevent excitotoxic ocular effects induced by intravitreal administration of glutamate agonists in rats was demonstrated. However, it is important to consider some limitations of this study.

Various animal models of ocular neuronal degeneration have been proposed. However, there is no reproducible and globally accepted model due to the complexity of the mechanisms involved in retinal neurotransmission. The excitotoxicity model used in this study is based on previous studies by P. de la Villa [10,11]. It involves the administration of NMDA and KA into rats’ vitreous cavity. Glutamate is the predominant neurotransmitter in the retinal network. NMDA administration primarily leads to the degeneration of amacrine cells and RGCs but does not directly affect the other cells in the retina, such as photoreceptor cells. Kainate receptors control a sodium channel that generates excitatory postsynaptic potentials when glutamate binds. Additionally, there are different receptor subunits in different cells of the retina. Therefore, experimental models do not fully reflect the natural conditions of ocular diseases in humans. Also, following the 3Rs rule, each formulation was administered to the minimum number of animals required. With a larger sample size, however, the results obtained could have had more powerful statistical significance. Finally, another aspect to consider is the duration of the study. Neurodegenerative ocular diseases have a prolonged course, requiring long-term studies to fully understand the complex implications over time.

## 3. Results

In order to evaluate retinal functionality, full-field ERG recordings were performed in dark-adapted conditions, after the MS administration and before the NMDA/KA-induced retinal lesion and 4 days after NMDA/KA-induced retinal lesion (Figure 1).

The strategy used for this purpose was developed based on previous validated works [10,11,17]. STR components were evaluated first in healthy rats’ eyes for flash intensities ranging from −5.5 to −4.4 log cd·s·m^−2^. With increasing stimulus intensity, both the pSTR and nSTR grew in amplitude. At −4.4 log cd·s·m^−2^, the positive and negative components showed their peak amplitudes at 250 ms and 150 ms, respectively. At the highest stimulus intensities, the ERG a- and b-waves reached their peak amplitudes at 250 ms and 800 ms, respectively.

There were significant reductions in the mean amplitude of pSTR (52% less), nSTR (49% less), the scotopic a-wave (38% less) and the scotopic b-wave (42% less) (ANOVA, Bonferroni’s test, *p* < 0.01 in all cases) when NMDA/KA was intravitreally injected (Figure 3). The NMDA/KA-mediated toxicity had no discernible effects on the implicit time of the pSRT or nSTR. However, a significant increase in the implicit time of both the a- and b-waves, approximately 7% in each case, was observed compared to untreated control rats (ANOVA, Bonferroni’s test, *p* < 0.05 for both). Prior to the formation of the NMDA/KA-induced lesion, the amplitude of oscillatory potentials (OPs) was notably higher, nearly 50%, compared to after the damage (ANOVA, Bonferroni’s test, *p* < 0.01). Remarkably, there were no observable differences in the implicit time in this comparison. These findings underscore the distinct effects on implicit times across different electroretinogram components.

ERG responsiveness was less attenuated by NMDA/KA in the F1-MS-, F2-MS- and F3-MS-treated animals than in the vehicle- or blank-MS-treated animals (Figure 3). Although the F1-MSs, F2-MSs and F3-MSs did not completely prevent the fall in retinal responsiveness upon NMDA/KA injection, they did reduce the damage considerably. Significant higher pSTR and nSTR amplitudes were recorded after NMDA/KA injection in the F1-MS-, F2-MS- and F3-MS-treated rats (97%, 73% and 54% increase compared to untreated eye response for pSTR and 87%, 91% and 37% increase for nSTR, respectively). No differences between NMDA/KA and blank-MS were recorded. (ANOVA, Bonferroni’s test, *p* < 0.05). In NMDA/KA-injected animals, the treatments also increased the scotopic a- and b-wave amplitudes to significantly higher values than those observed in NMDA/KA-administered control rats (24%, 18% and 22% increase for a-waves with F1-MSs, F2-MSs and F3-MSs, respectively, and 34%, 6% and 11% for b-waves, respectively; ANOVA, Bonferroni’s test, *p* < 0.05 in both cases). The influence of the formulations on the a-wave implicit time in NMDA/KA-injected rats showed minimal impacts. However, F1-MSs, F2-MSs, and F3-MSs administration resulted in a significant reduction in the b-wave’s implicit time in NMDA/KA-damaged rats, with reductions of 1.5%, 1.8%, and 2.4%, respectively (ANOVA, Bonferroni’s test, *p* < 0.05). Furthermore, treatment with F1-MSs, F2-MSs, and F3-MSs significantly alleviated the detrimental effects of NMDA/KA on the scotopic oscillatory potentials (OPs). While the treatments did not completely reverse the decrease in OP amplitude caused by NMDA/KA, the scotopic OP amplitudes after the NMDA/KA-induced lesion were significantly higher, with increases of 18% and 8% with F1-MSs and F2-MSs, respectively, and no significant differences with F3-MSs (ANOVA, Bonferroni’s test, *p* < 0.05).

Following intravitreal NMDA/KA delivery, the number of RGCs decreased. To verify the correlation between retinal function impairment and the number of surviving RGCs, we performed immunostaining for Brn3a, which revealed a distinct nuclear localization signal within viable retinal ganglion cells (RGCs), consistent with its established role as a transcription factor [26]. This staining pattern corroborates earlier observations [27], which indicated that RGC density follows a gradient across the retina. Specifically, a higher density was observed in the central retina, precisely at 2 mm from the optic nerve head. As one moved towards the periphery, approximately 4 mm from the optic nerve head, there was a gradual decrease in RGC density. Notably, the dorsal quadrant of the retina, located 2 mm from the optic nerve head, exhibited the highest RGC density.

The injection of NMDA/KA into the vitreous chamber induced a global decrease in the density of RGCs compared to the non-injected control eyes (30% less, Figure 4), which can be readily observed in the retinal micrographs (Figure 5).

To further confirm the protective effect of the MS formulations on RGCs, the density of Brn3a-positive cells in the retinal ganglion cells layer (RGCL) of retinal flat-mounts was compared between the vehicle-, Dxm-vehicle- and F1-, F2- and F3-MS-vehicle-treated rats. Overall, the RGC survival, expressed as the percentage of the average RGC density in the non-injected control rats, was significantly higher (Student’s *t* test, *p* < 0.05) in all MS-treated animals after the NMDA/KA injection (Figure 4 and Figure 5). When the RGC sections were taken into account, a significant (two-way ANOVA, *p* < 0.001) protective effect of the MS formulations was detected in the central areas of the retina (2 mm from the optic nerve head) but only a slight effect in the peripheral areas (4 mm from the optic disc). The average RGC density was higher in Dxm-vehicle- and F1- and F2-MS-vehicle-treated group than in the F3-MS-vehicle-treated group.

Finally, the effect of the different treatments on retinal photoreceptors and on the inner retina was studied. Photomicrographs of retinal sections stained for cone arrestin, cone blue opsin and cell nuclei demonstrated no differences between the untreated and Dxm-vehicle-, F1-, F2- or F3-MS-vehicle-treated groups (Figure 6A). However, in the inner retina, the photomicrographs of retinal sections stained for the Brn3a transcription factor, choline acetyltransferase and cell nuclei showed a disorganization of the inner plexiform layer and retinal ganglion cells in Dxm-vehicle- and F3-MS-vehicle-treated groups, but not after treatment with vehicle or F1- or F2-MSs.

## 4. Discussion

The major challenge in the field of retinal excitotoxic neurodegenerative diseases is halting or attenuating vision loss due to neuronal cell death [28]. Glutamate excitotoxicity plays a major role in the loss of retinal ganglion cells in glaucoma. The toxic effects of glutamate on RGCs are mediated by the overstimulation of N-methyl-D-aspartate receptors [29]. The immunohistochemical examination of the rat retina unveiled the localization of NMDA receptor subunits. These subunits were prominently identified on retinal ganglion cells (RGCs) and displaced amacrine cells situated within the ganglion cell layer. Furthermore, a distinct subset of amacrine cells in the inner nuclear layer exhibited the presence of NMDA receptor subunits. However, there is no evidence of functional NMDA receptors on photoreceptor cells [30]. Consequently, NMDA administration primarily results in the degeneration of amacrine cells and RGCs, but does not directly affect other cells in the retina, such as photoreceptor cells. For this reason, NMDA has been commonly used as an experimental tool to induce glutamate-induced excitotoxic retinal ganglion cell death. Upon injecting NMDA into the vitreous chamber of Sprague Dawley rats, a marked and acute decline in the number of retinal ganglion cells (RGCs) was observed, reaching the peak of cell death 48 **h** post-administration. The degeneration of RGCs, and presumably amacrine cells could account for the alterations seen in the electroretinogram (ERG) results. This encompasses changes in parameters such as the pSTR, nSTR, b-wave, and oscillatory potentials (OPs).

NMDA receptor antagonists have been proposed as a pharmacological approach for the treatment of degenerative diseases where excitotoxicity is involved. However, a complete NMDA receptor blockade results in neurotoxicity [12]. An interesting alternative would be the use of antiapoptotic and antioxidant compounds to provide neuroprotection to RGCs.

Dxm is widely used in ocular inflammatory diseases [18]. Despite its clinical relevance, due to its hormone characteristics, there is a pleiotropic drug effect that is currently not fully understood. For example, over the last decade, studies have reported the effects of Dxm on lipid peroxidation and nitric oxide levels [31], in the suppression of the JAK2/STAT3 pathway by inhibiting interleukin-6, TNF-α or ICAM-1 mRNA expression [32] and in the reduction of cell migration involving the ERK and ATK pathways and the target factor CYR61 [33].

Calvo and coworkers [17] demonstrated that drugs with a carboxylic group bound to an aromatic ring (such as salicylate, sulindac, indomethacin, ibuprofen, or R-flurbiprofen) are able to partially depolarize mitochondria and inhibit NMDA-induced mitochondrial Ca^2+^ uptake via the endogenous mitochondrial Ca^2+^ uniporter, without affecting [Ca^2+^]_cyt_ increases. This resulted in less NMDA-induced apoptosis in neurons. Here, based on the structural similarity of Dxm, which has an accessible hydroxyacetyl group near an aromatic structure and a suitable partition coefficient (log *p* = 2.03 ± 0.6) to reach the endogenous mitochondrial Ca^2+^ uniporter, we have hypothesized that Dxm has the same behavior in RGCs.

Excitotoxicity is a chronic pathological phenomenon and pharmacological interventions must ensure the optimal therapeutic concentration in the retina for as long as it is necessary. In that sense, due to the difficulty of access and ruling out systemic administration due to the risk of side effects in the long-term, a strategy based on drugs administered intravitreally would be required for therapies involving repeated intraocular injections [25]. Drug delivery systems may address this issue, as they are able to release the drug over a long period of time [34]. In particular, biodegradable microspheres (<40 μm) can be administered intravitreally in the form of a suspension using conventional needles. Microspheres are able to provide controlled release of active substances and are good candidates for use in personalized medicine, as different amounts of particles can be administered depending on the patient’s needs.

Our results from an acute retinal degeneration model in rats demonstrated that the intravitreal administration of MSs loaded with dexamethasone is able to play a protective role in RGC viability against excitotoxic injury, similar to an intravitreal injection of a high dose of dexamethasone. In this study, the injury was caused by the intravitreal administration of NMDA and KA at low concentrations to prevent total blindness [29]. Here, the maximum deleterious effect of the administered NMDA/KA was achieved after 7 days; however, the pathologic effect of excess glutamate is chronic and the optimal concentration of dexamethasone must be maintained over time. Biodegradable microspheres that can control the release of therapeutic amounts of dexamethasone over a prolonged period, as proposed here, can be an interesting approach for these illnesses.

The ideal release profile of dexamethasone is a low initial burst but enough to achieve the proposed minimum concentration required for the excitotoxicity blockage effect, followed by sustained delivery to maintain the therapeutic concentration in the vitreous humor for at least seven days. According to the results obtained from the in vitro release studies, microspheres from the 20–40 µm fraction can achieve this goal from the outset to seven days later. After the first 4 h, all MS formulations released at least 4 µg of dexamethasone. The average in vitro release rate during the first seven days was 7.6 µg Dxm/mg MSs/day for F1-MSs and F2-MSs, and 5.5 µg Dxm/mg MSs/day for F3-MSs [21].

Our MSs were not only able to maintain retinal cell functionality (Figure 3), but also their integrity and morphology without significant disruptions to the retinal cellular layers (Figure 6). The RGC degeneration correlated well with the diminished amplitudes of the pSTR, nSTR, b-wave and oscillatory potentials on the electroretinograms under scotopic conditions. As several reports suggest, in rodents, the ERG response to a very dim light stimulus depends on inner retina function, specifically RGCs [35,36]. The pSTR and nSTR are dominated by the amacrine or ganglion cell responses. RGC death due to retina alterations modifies or eliminates both responses; the nSTR is absent from eyes in which ganglion cells have been abolished because of, e.g., laser-induced ocular hypertension. The b wave reflects the activity of rod-driven bipolar cells and the photoreceptors’ output to the proximal retina and its amplitude depend on the integrity of signal transmission within the retina [36]. Subnormal b-waves with normal implicit times, as observed here, have been noted in specific degenerative pathologies such as macular degeneration [37]. Furthermore, RGCs and amacrine cells are also involved in OP changes and decreased amplitudes are significantly correlated with pathophysiological excitotoxic processes [38]. Pretreatment with dexamethasone in a controlled release system has proved to be effective at increasing RGC survival and retinal function. However, the molecular mechanism has not been fully characterized. It has been proposed that there is a blockade of the transcriptional factor Bax while it is translocating through the mitochondria, which could prevent cytochrome C release [39]. Also, the stimulation of survival pathways, including the PI3K and MAPK (p38, ERK1/2) pathways [40], or the inactivation of reactive microglia and macrophages [41] might be involved. When vitamin E is included in the MSs, it also can exert antioxidant activities on the retinal cells, preventing the negative effects of the oxidative stress [23].

Regarding integrity, as it was previously noticed by [42], a greater reduction in RGCs was observed in the peripheral retina (areas 2, 4, 6 and 8), while surviving RGCs were distributed more densely in the vicinity of the optic disc (areas 1, 3, 5 and 7 from the optic disc). The results showed that more RGCs died close to the optic disc and the number was gradually reduced as we move to periphery.

By using antibodies, we were able to analyze the organization of the retina layers after the degeneration of RGCs. It must be noted that morphological changes, and finally functionality as well, took place when the dexamethasone concentrations were higher than the antiapoptotic concentration or the drug concentration was not achieve or maintain during the excitotoxic process. As shown in Figure 6, retinal photoreceptors were not compromised but in the inner retina of Dxm-vehicle- and F3-MS-veh-treated eyes, a disorganization of the inner plexiform layer and retinal ganglion cells was observed. This fact, linked with the abnormal thickness, is considered to be the first histological alteration after excitotoxicity and indicates changes in the phenotypic expression. In F3-Ms-vehicle treatment findings suggest that the Dxm concentration administered is not enough to prevent degeneration and disorganization of the retinal layer. The administration of high doses of dexamethasone could possibly induce the apoptosis of RGCs as a result of overactivation of the nuclear receptor. High doses of steroids have been correlated with damage to the optic nerve, retinal epithelium damage and local ion flow changes [43].

## 5. Conclusions

We clearly demonstrated that some of the microparticles developed (F1-MSs and F2-MSs) can act as a controlled release system that provides sustained and low concentrations of dexamethasone, which can disrupt the excitotoxic mechanisms induced by NMDA/KA. They showed the ability to maintain normal electroretinal activity and provided significant protection to RGCs near the optic nerve head in an ocular rat model of acute excitotoxicity injury. Through this proof of concept, we demonstrated that MSs loaded with dexamethasone could be a useful strategy to avoid the progression of degenerative ocular diseases.

## Figures and Tables

**Figure 1 pharmaceutics-16-00406-f001:**
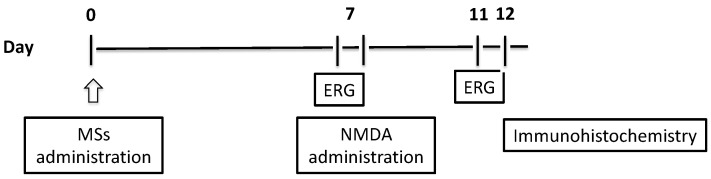
Experimental procedure timeline.

**Figure 2 pharmaceutics-16-00406-f002:**
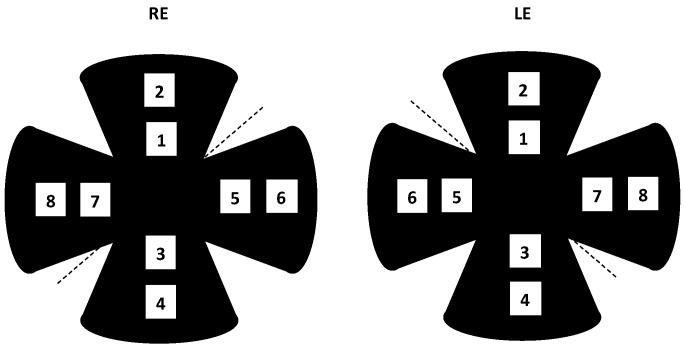
Scheme of retinal areas used to calculate the average cell density. Areas 1, 3, 5 and 7 are 2 mm from the head of the optic nerve, while areas 2, 4, 6 and 8 are in the periphery at 4 mm from the head of the optic nerve.

**Figure 3 pharmaceutics-16-00406-f003:**
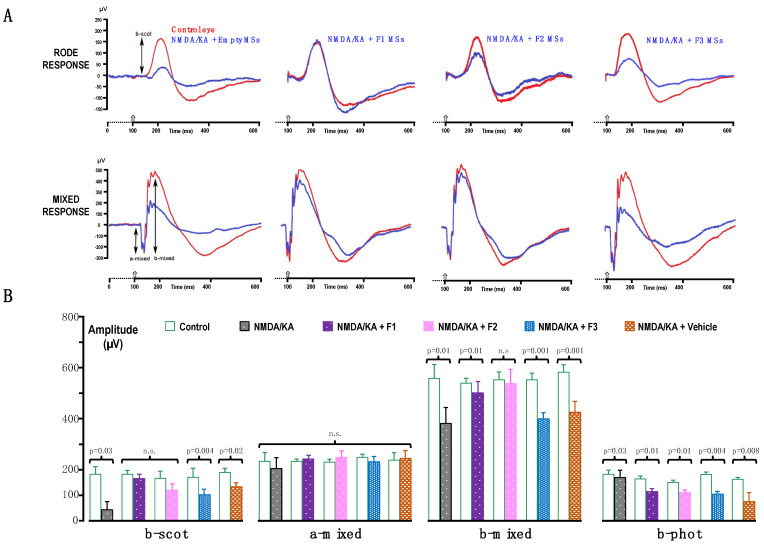
Effect of the different treatments on electroretinogram (ERG) wave amplitudes. (**A**). Representative ERG traces of the rod and the mixed scotopic responses recorded from single animals treated with empty microspheres and loaded microspheres (F1-MSs, F2-MSs and F3-MSs). The rod response was induced by a −2 log cd·s·m^−2^ flash; the mixed response was induced by a 1.5 log cd·s·m^−2^ flash. Arrows show the measurement for the a-wave and b-wave amplitudes. (**B**). Histogram representation of the ERG wave amplitudes averaged from 6 animals from each experimental group. Bars correspond to the mean data (mean ± SD) of the b-wave amplitudes measured from the rod response (b-scot), and the a-wave and b-wave amplitudes measured from the mixed response (a-mixed, b-mixed) for the control, NMDA/KA and treated eyes of each experimental group. Statistically significant differences between both eyes are indicated above the bars. n.s.: *p* > 0.05.

**Figure 4 pharmaceutics-16-00406-f004:**
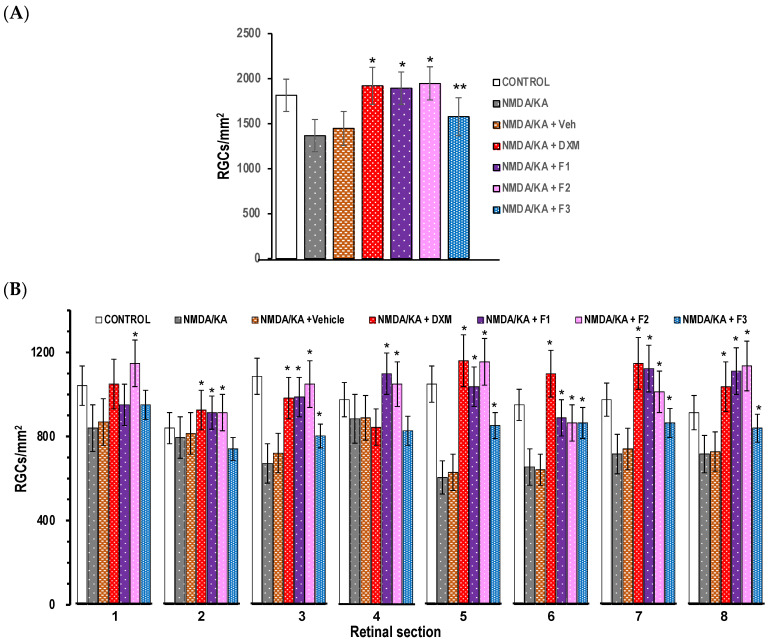
(**A**) Density of retinal ganglion cells (RGCs/mm^2^) in each group (* *p* < 0.01, ** *p* < 0.05). (**B**) Density of retinal ganglion cells (RGCs/mm^2^), averaged from each retinal area (1–8, see Figure 2). The densities for animal groups injected intraocularly with 4 μL of a solution containing NMDA/KA (1 mM and 0.3 mM, respectively) are shown. The histogram bars correspond to the values from the control group (NMDA/KA), group preinjected with 0.1% *w/v* sodium hyaluronidate (vehicle), Dxm in vehicle and F1-MSs, F2-MSs and F3-MSs in vehicle. The results are displayed as mean ± SD. * *p* < 0.01.

**Figure 5 pharmaceutics-16-00406-f005:**
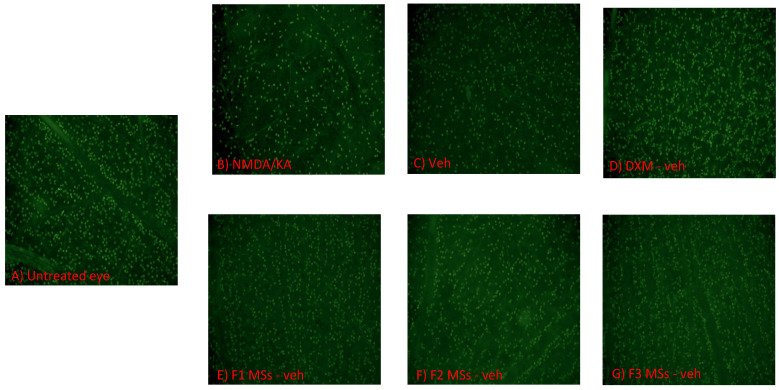
Photomicrographs of wholemount retinas stained for Brn3a transcription factor expressed in retinal ganglion cells. The sample images are from animals from each of the experimental groups shown in Table 2. The figure shows the RGC viability in area 3 of the retinas without (control) (**A**) and after intravitreal injection of 4 μL of a solution containing NMDA/KA (1 mM and 0.3 mM) (**B**). Images are from the experimental groups pretreated with sodium hyaluronidate 0.1% *w*/*v* (vehicle) (**C**), Dxm in vehicle (**D**), or F1-MSs (**E**), F2-MSs (**F**) or F3-MSs (**G**) in vehicle, are also shown.

**Figure 6 pharmaceutics-16-00406-f006:**
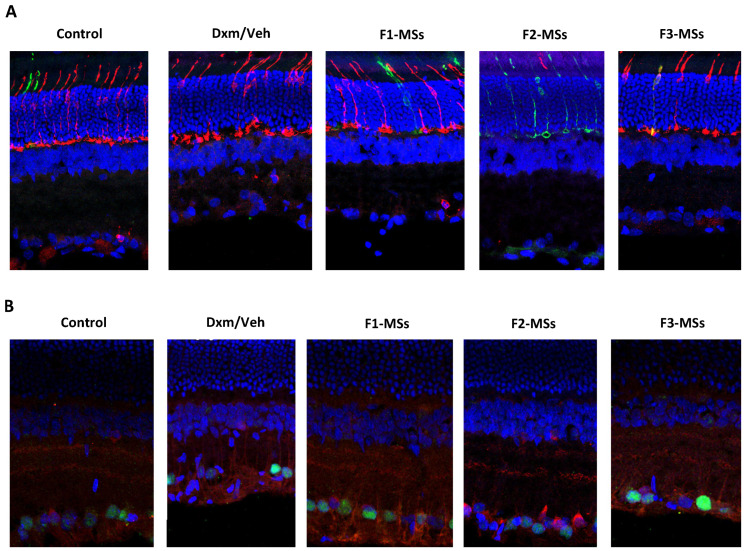
Effect of the different treatments on retinal photoreceptors and on the inner retina. (**A**). Photomicrographs of retinal sections stained for cone arrestin (red), cone blue opsin (green) and cell nuclei (blue) from a control eye and eyes injected with Dxm in vehicle, F1-MSs, F2-MSs or F3-MSs in vehicle. No differences among the eyes were observed. (**B**). Photomicrographs of inner retinal sections stained for the Brn3a transcription factor (green), choline acetyltransferase (red) and cell nuclei (blue) from control and eyes injected with Dxm in vehicle, F1-MSs, F2-MSs or F3-MSs in vehicle. A disorganization of the inner plexiform layer and retinal ganglion cells was observed in the Dxm in vehicle and F3-MSs groups.

**Table 1 pharmaceutics-16-00406-t001:** Summary of the main characteristics for each MS formulation. Composition, drug and human serum albumin loading, burst effect after 4 h and dexamethasone released after 7 days are included.

MSs	Composition				Loading (μg/mg MSs) (a) Dxm (b) HSA	Dxm Burst Release after 4 h (%)	Dxm Released after 7 Days (μg/mg MSs)
	PLGA	Dxm	Vit. E	HSA			
Blank	200 mg	-	-	-	-	-	-
F1	200 mg	20 mg	-	-	(a) 90.5 ± 11.3	3.50 ± 0.79	36.26 ± 3.27
F2	200 mg	20 mg	20 μg	-	(a) 80.9 ± 14.1	4.60 ± 1.64	28.31 ± 2.83
F3	200 mg	20 mg	20 μg	20 μg	(a) 84.8 ± 12.7 (b) 0.113 ± 0.001	1.36 ± 0.25	12.72 ± 1.31

**Table 2 pharmaceutics-16-00406-t002:** Details of the experimental groups.

Group	G1	G2	G3	G4	G5	G6
**Treatment**	No treatment (NMDA/KA)	Hyaluronic acid (0.1% *w*/*v*) (vehicle)	Dxm and hyaluronic acid (0.1% *w*/*v*)	F1-MSs in hyaluronic acid (0.1% *w*/*v*)	F2-MSs in hyaluronic acid (0.1% *w*/*v*)	F3-MSs in hyaluronic acid (0.1% *w*/*v*)

## Data Availability

The data can be shared up on request.

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
