# Peer review of "Useful Role of a New Generation of Dexamethasone, Vitamin E and Human Serum Albumin Microparticles in the Prevention of Excitotoxicity Injury in Retinal Ocular Diseases"

_pharmaceutics, 2024, doi:10.3390/pharmaceutics16030406_

Round 1
Reviewer 1 Report
Comments and Suggestions for Authors
The authors of the manuscript _ ID: pharmaceutics-2865054 titled “Useful role of a new generation of dexamethasone, vitamin E and human serum albumin microparticles in the prevention of excitotoxicity injury on retinal ocular diseases”, presented research on the role of dexamethasone in neuroprotection. For this purpose, proprietary controlled-release systems have been developed consisting of microparticles containing dexamethasone with or without vitamin E and human serum albumin (HSA). These microparticulate systems showed in an ocular rat model the ability to maintain normal electroretinal activity and promoted significant protection of RGCs . The authors demonstrated that dexamethasone could be a useful anti-inflammatory agent to avoid a degenerative ocular disease’s progression.
A well-prepared and interesting manuscript with great clinical potential. The introduction is comprehensive and supported by appropriate literature.
Materials and methods: Line 95-96 add the company and country of the product: Poly (D,L-Lactide-co-glycolide) acid and Vitamin E.The authors conducted research using many research tools including electroretinogram,iImmunohistochemistry, confocal microscopy and quantification of surviving RGCs.
he description of Figure 4A is hard to see, please correct it. The discussion is well conducted. Some authors are quoted in the discussion (by name and year of publication of the article), but they are not included in the reference list. Add what limitations the study has.
The authors put a lot of work into it, thank you.
Author Response
Dear reviewer, thanks for your support and nice words.
Now, materials are completely described.
Figure 4 has been modified in order to improve clarity. Also, colors in figure 3 has been adapted to facilitate legibility.
Bibliography has been checked and completed.
A specific paragraph for research limitations have been added.
Limitations
In this study, the ability of PLGA microspheres loaded with Dxm to prevent excitotoxic ocular effect induced by intravitreal administration of glutamate agonists in rats has been demonstrated. However, it is important to consider some limitations of this study, which should be mentioned.
Various animal models of ocular neuronal degeneration have been proposed. However, there is no reproducible and globally accepted model due to the complexity of the mechanisms involved in retinal neurotransmission. The excitotoxicity model used in this study is based on previous studies by P. de la Villa (11-12). It involves the administration of NMDA and KA into rats’ vitreous. Glutamate is the predominant neurotransmitter in the retinal network. NMDA administration primarily leads to the degeneration of amacrine cells and RGCs but does not directly affect other cells in the retina, such as photoreceptors. Kainate receptors control a sodium channel that generates excitatory postsynaptic potentials when glutamate binds. Additionally, there are different receptor subunits in different cells of the retina. Therefore, experimental models do not fully reflect the natural conditions of ocular diseases in humans.
Also, following the 3Rs rule, each formulation has been administered to the minimum number of animals required. With a larger one, however, the results obtained would have had more powerful statistical significance.
Finally, another aspect to consider is the duration of the study. Neurodegenerative ocular diseases have a prolonged course, requiring long-term studies to fully understand the complex implications over time.
Thanks,
Reviewer 2 Report
Comments and Suggestions for Authors
pharmaceutics-2865054
Useful role of a new generation of dexamethasone, vitamin E and human serum albumin microparticles in the prevention of excitotoxicity injury on retinal ocular diseases
The manuscript by Villanueva et al. described the development of microparticles containing dexamethasone for the prevention of excitotoxicity injury on retinal ocular diseases. The specific comments are as follows. Considering them, this manuscript is unsuitable for publication in Pharmaceutics.
1. The rationale of this work is unclear. The authors did not clarify the research gap, why it was necessary to develop microparticles containing dexamethasone, and why vitamin E and HSA were selected.
2. Injection of microparticles to the eyes can cause irritation and discomfort to patients. One should use solution forms or nanoparticles with much smaller sizes.
3. The authors should design and evaluate more formulations to obtain the optimized formulation. In this study, there are only 3 formulations (except blank PLGA): Dxm PLGA, Dxm VitE PLGA, and Dxm VitE HAS PLGA microparticles.
4. Generally, a formulation should be well characterized in vitro before being applied to animal studies to avoid unnecessary animal experiments. The authors should perform in vitro characterization of the formulations.
5. The quality of Figures 2, 3, 4, and 6 is low.
6. The authors did not show data to support the “controlled release” of the drug.
Comments on the Quality of English Language
Minor editing of English language required
Author Response
Dear reviewer, thanks for your comments.
Now, figures 3 and 4 has been modified in order to improve clarity.
Also, please take into account that this manuscript is the second stage of our project. As it is indicated, the necessity of the research, the objectives, the possibilities to achieve them and the original design can be found here: Rodriguez Villanueva, J., et al., Optimising the controlled release of dexamethasone from a new generation of PLGA-based microspheres intended for intravitreal administration. Eur J Pharm Sci, 2016. 92: p. 287-97. Also, in the same work we explain the evaluation done to achieve an optimized formulation, develop a complete in-vitro characterization of the MS and analyze the controlled release of the drug. As a result, we presented there an efficient and reproducible method for encapsulating Dxm in biodegradable PLGA (50:50) microspheres that are viable for intravitreal injection with 30 G needles. MSs prepared; highlighting those containing Dxm, vitamin E and HSA, showed a good Dxm controlled release and cellular behavior (similar to dexamethasone solutions). For these reasons, our MSs were good candidate for further in vivo studies, that are the main objective of the work presented here.
Reviewer 3 Report
Comments and Suggestions for Authors
I read the paper entitled " Useful role of a new generation of dexamethasone, vitamin E and human serum albumin microparticles in the prevention of 3 excitotoxicity injury on retinal ocular diseases" done by Villanueva et al. The manuscript looks interesting and this is a worthwhile subject. In addition to the well-organized research paper, the author's presentation of the results and the drawn figures is impressive. However, some minor modification need to be done regarding this paper as following. The following point should be added to the revised manuscript and changed.
Please modify this in the entire manuscript.
Figure 4 (A and B) are not clear. I suggest the author replace them.
Add statistical analyses section to your manuscript. What consider as statically difference.
Provide the defining for example for n.s. , p=0.001, ….. under each Figure.
Author Response
Dear reviewer, thanks for your words.
Figure 4 has been modified in order to improve clarity.
A specific paragraph for statistical analysis have been added.
Statistical analysis
Previously [22], the statistical significance of the different parameters (yield of production, loading efficiency and particle size) among different batches of microspheres was tested by a one-way analysis of variance (ANOVA) with the Pairwise Multiple Comparison Procedures. Dxm-loaded microspheres' release profiles were compared with the evaluation of the similarity factor f2. Differences were considered to be significant at a level of p < 0.05.
Here, statistical analyses were performed using SPSS 18.0 software (IBM Armonk, NY, USA) and GraphPad Instat® 3 for Windows® (GraphPad Software, San Diego, CA). Descriptive statistics were calculated, and the normality of the distribution of the data was examined. A two-way repeated-measures ANOVA was performed to evaluate the effects of the treatment (F1-F3-MSs + NMDA/KA vs. Dxm, Blank-MSs or Vehicle + NMDA/KA or NMDA/KA) on ERG responses. A two-way ANOVA was performed to evaluate differences in the mean density of RGCs between the experimental groups (NMDA/KA, Vehicle, Blank-MSs, F1-MSs, F2-MSs and F3-MSs) at the distinct eccentricities. When a 0.05 level of significance was found, post-hoc pairwise comparisons using Bonferroni’s test were made. P values less than 0.05 were considered statistically significant. Data were plotted as the mean ± standard deviation (mean ± SD). When needed, the coefficient of variation (CV) is presented as a percentage SD/arithmetic mean ratio. Microsoft Excel (2019) were used to create bar graphs. ANOVA test was used to assess the significant differences among various experimental groups.
A definition is now included under each figure.
Thanks,
Round 2
Reviewer 2 Report
Comments and Suggestions for Authors
The manuscript was revised accordingly. Below are some issues to consider in this revised manuscript.
1. The decimal separators should be ".". Please check and correct (e.g., see Figure 3B).
2. Figure 4B, retinal section 3: It seems to have an extra "*".
3. The Introduction needs to be improved to clarify the novelty and contribution of this study.
4. Please mention and discuss the injectability of the formulations.
5. Please discuss the advantages of the MS in comparison with nanosystems for intravitreal delivery.
Author Response
Dear reviewer, thanks for your comments.
Figures 3 and 4 have been checked and corrected.
Also, introduction has been improved to include advantages of MS in comparison with nanosystems for intravitreal delivery and the novelty of the work presented here; finally, methodology and discussion have been updated to deepen into the injectability and highlight the contribution of the results achieved to the field.
Round 3
Reviewer 2 Report
Comments and Suggestions for Authors
The revised manuscript can be accepted for publication.